

# Mass mortality event of the giant barrel sponge *Xestospongia* sp.: population dynamics and size distribution in Koh Phangan, Gulf of Thailand

Jasmin S. Mueller[1,2], Paul-Jannis Grammel[1,2], Nicolas Bill[1], Sven Rohde[1] and Peter J. Schupp[1,3]

[1] Institute for Chemistry and Biology of the Marine Environment (ICBM), Carl von Ossietzky University Oldenburg, Wilhelmshaven, Germany
[2] Center for Oceanic Research and Education (CORE sea), Chaloklum, Koh Phangan, Surat Thani, Thailand
[3] Helmholtz Institute for Functional Marine Biodiversity (HIFMB), Carl von Ossietzky University Oldenburg, Oldenburg, Germany

Corresponding author
Peter J. Schupp,
peter.schupp@uni-oldenburg.de

## ABSTRACT

Marine sponges are prominent organisms of the benthic coral reef fauna, providing important ecosystem services. While there have been increasing reports that sponges are becoming one of the dominant benthic organisms in some locations and ecoregions (*e.g.* Caribbean), they can be impacted by changing environmental conditions. This study presents the first documentation of a mass mortality event of the barrel sponge *Xestospongia* sp. in the lower Gulf of Thailand and its consequences on population dynamics and size distribution. Two anthropogenic impacted reefs (Haad Khom and Mae Haad) of the island Koh Phangan and two anthropogenic non-impacted reefs of the islands Koh Yippon and Hin Yippon within the Mu Ko Ang Thong Marine National Park were surveyed in the years 2015 and 2016. The results showed a strong shift in population densities at Koh Phangan. Fatal "bleaching" ending up in mass mortality was observed for these reefs in 2015. *Xestospongia* sp. abundance decreased from 2015 to 2016 by 80.6% at Haad Khom and by 98.4% at Mae Haad. Sponges of all sizes were affected, and mortality occurred regardless of the survey depth (4 and 6 m). However, *Xestospongia* population densities in the Marine Park were at a constant level during the surveys. The abundances in 2015 were 65% higher at the Marine Park than at Koh Phangan and 92% higher in 2016. The most likely causes of the mass mortality event was a local harmful algal bloom event, pathogens, undetected local higher water temperatures, or a combination of these factors, whereas sea surface temperature analyses showed no marine heatwave during the observed mass mortality event in 2015. Considering the ecological importance of sponges such as *Xestospongia* sp., long-term monitoring of reefs and their environmental parameters should be implemented to prevent such mass die-offs.

## INTRODUCTION

Various marine sponges play an important role as engineers of coral reefs (*Bell, 2008*; *Rix et al., 2016*), have a positive impact on the habitat complexity and the local biodiversity (*Maldonado et al., 2016*), provide home for invertebrate taxa (*Wulff, 2006*), and function as a food source (*Coppock et al., 2022*; *Engel & Pawlik, 2005*). Due to their ability to filter large amounts of seawater (*Reiswig, 1974*), they have been recognized as an important component of benthic-pelagic coupling. In recent years there has been increasing evidence that sponges play an important role in biochemical cycles (*Pawlik & McMurray, 2020*), in particular the uptake and transformation of dissolved organic matter (DOM) to particular organic matter (POM) *via* the so-called sponge-loop (*de Goeij et al., 2013*; *Rix et al., 2016*; *Schläppy et al., 2010*).

The barrel sponge *Xestospongia* sp., a genus of the family Petrosiidae, is a key species in tropical, shallow coral reefs (*McMurray, Finelli & Pawlik, 2015*). This sponge is an ecological important member of the reef community, due to its large size, its importance as a habitat engineer, and its function as a big seawater filterer (*McMurray, Pawlik & Finelli, 2014*; *McMurray, Henkel & Pawlik, 2010*). Until recently, the most abundant species of the genus *Xestospongia* were identified as *X. muta* in the Caribbean and *X. testudinaria* in the Indo-Pacific. However, recent genetic studies documented that these different *Xestospongia* species are part of a more diverse species complex (*Evans et al., 2021*; *Swierts et al., 2017*). Morphologically similar species occur sympatrically and might feature differences in their physiology. Some individuals of the Caribbean reefs may be older than 2,300 years and can grow several meters in height and diameter (*Gammill, 1997*; *Humann & Deloach, 1992*; *McMurray, Blum & Pawlik, 2008*). *McGrath et al. (2018)* found contrary to McMurray twice as high growth rates for *Xestospongia* spp. from Wakatobi National Marine Park, Indonesia, compared to the Caribbean counterparts.

The main habitat-forming organisms, the scleractinian corals, are threatened worldwide by multiple stressors such as rising water temperatures, ocean acidification, and pollution (*Mollica et al., 2018*; *Mueller et al., 2022*; *Prada et al., 2017*). With the decline of hard coral cover in shallow waters, coral reefs can undergo phase shifts from coral-dominated states to reefs dominated by algae or other invertebrates, including sponges (*Bell, Micaroni & Strano, 2022*; *Bell et al., 2013*; *de Bakker et al., 2017*; *Dinsdale & Rohwer, 2011*; *Maliao, Turingan & Lin, 2008*; *Reverter et al., 2022*). Reefs dominated by sponges have mainly been reported from the Caribbean but also from certain locations in the Indo-Pacific (*Bell et al., 2013*; *Maliao, Turingan & Lin, 2008*; *Powell et al., 2010*; *Reverter et al., 2021*).

*Xestospongia* spp. have been reported to undergo cyclic bleaching like reef-building corals (*McMurray et al., 2011*). During this time, the sponges lose their reddish-brown coloration due to the loss of their photosynthetic symbionts and appear creamy-white (*Erwin & Thacker, 2007*; *Vicente, 1990*). Unlike corals, the affected *Xestospongia* spp. can survive long-term bleaching without mortality. According to *McMurray et al. (2011)*, this fact leads to the assumption that the host benefits just a little or not from its photosynthetic symbionts.

With increasing sponge research and benthic monitoring in the last decades, also more bleaching events, diseases, and mortality events were reported for sponges around the world (*Angermeier et al., 2011*; *Belikov et al., 2018*; *Luter, Whalan & Webster, 2010*; *McMurray et al., 2011*; *Stevely et al., 2011*; *Webster et al., 2008*). Sponge mortality is often caused by fatal bleaching with massive tissue loss due to diseases (*Cowart et al., 2006*) or other environmental stressors such as changing environmental conditions and harmful algal blooms (HABs) (*Bauman et al., 2010*; *Cebrian et al., 2011*).

Our knowledge of *Xestospongia* spp. demography and size distribution in tropical regions like the Indo-Pacific is still limited. Further research is needed to fill this gap, especially since sponges generally could play an increasing role for tropical reefs in the future. This study aimed to monitor the population dynamics and the size distribution of *Xestospongia* sp. in the lower Gulf of Thailand. When the bleaching and massive die-off occurred during the monitoring, the aims were complemented with the documentation of the sponge health status to identify the population consequences of this event.

## MATERIALS AND METHODS

The surveys were conducted under a Memorandum of Understanding with the Department of Marine and Coastal Resources Thailand and the Phuket Marine Biological Center, and with a research permit with permit number B8425313.

Surveys were performed at different time points in March and April of the years 2015 and 2016. To determine the size and age structure of the *Xestospongia* populations at the study sites, surveys were carried out in March 2015 before the mortality event took place, once the mass mortality was recognized during the ongoing mortality event in April 2015, and during the next field season in March and April 2016. All data were collected by snorkeling.

### Study sites

Four coral reefs were chosen in the shallow waters of the lower Gulf of Thailand. Two of them were located on the island of Koh Phangan, and the other two were in the Mu Ko Ang Thong Marine National Park (Fig. 1). By choosing these study sites, the observed reefs were divided into reefs with different human impact levels: anthropogenic impacted (Koh Phangan), hereafter abbreviated with AI, and anthropogenic non-impacted (Marine National Park), hereafter abbreviated with ANI.

The investigated reefs of Koh Phangan were located in the northern and northwestern part of the island at the beaches Haad Khom (AI) (9°47′5″N, 100°00′53″E) and Mae Haad (AI) (9°47′30.9″N, 99°58′35.3″E) (Fig. 1B). Both were impacted by tourism. Several bungalows and bars are located at Mae Haad (AI), snorkelers got easily to the reef on foot, and fishing was recognized in front of the reef during our surveys. Haad Khom (AI) was less touristic, and snorkelers just got to the reef by boat. The reef of Mae Haad (AI) was during the surveys less degraded than the reef of Haad Khom (AI), which showed lower live coral cover and higher cover of dead coral overgrown by filamentous algae (JS Mueller, P-J Grammel and E Schoenig, 2015, 2016, personal communication). In contrast to this were the reefs around the small islands Koh Yippon (ANI) and Hin Yippon (ANI)

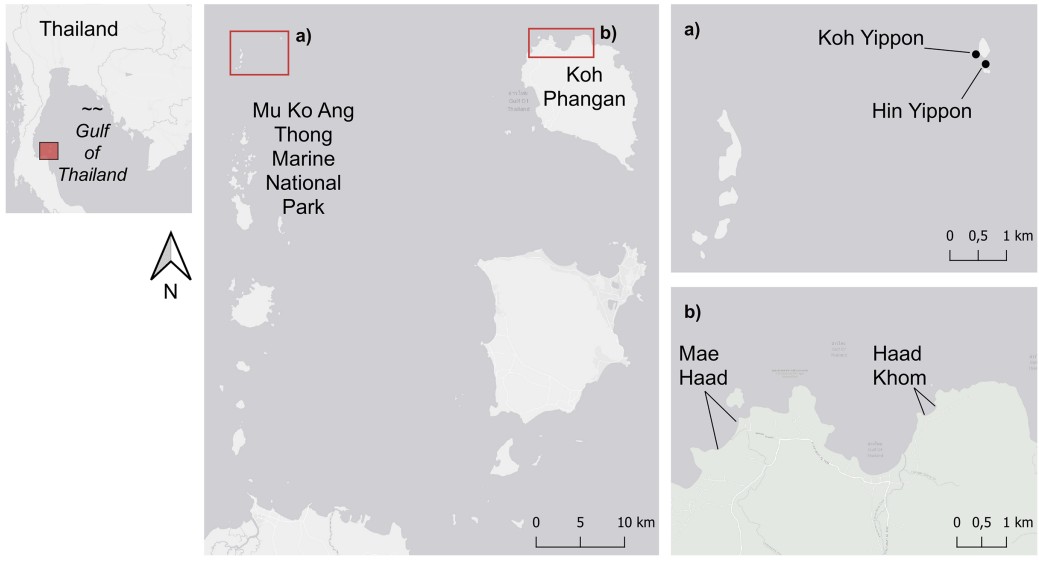

**Figure 1 Map of the study sites.** The study sites within the Mu Ko Ang Thong Marine National Park (ANI) are shown in (A), and (B) shows the two study sites on the island of Koh Phangan (AI). Figure source credit: The visualization of this map was done by using the ERSI ArcGIS® World Light Gray Base (basemap) version 7.2. Sources: Esri, HERE, Garmin, © OpenStreetMap contributors, and the GIS user community.

(9°47′54″N, 99°42′50″E) (Fig. 1A). The two islands are in the northern part of the Mu Ko Ang Thong Marine National Park and are approximately 35 km to the west of Koh Phangan. The observed coral reefs are located at the south-western part of Koh Yippon (ANI) and the northern part of Hin Yippon (ANI). These sites are exposed to little human influence due to the Marine National Park, founded in 1980 (*White, 1991*, p. 348). Tourism at the Marine National Park is limited, and diving activities are just permitted for a few parts of the park. The two islands are uninhabited and therefore less anthropogenically impacted than the sites on Koh Phangan.

## Field survey methods

All reefs were surveyed in water depths of 4 and 6 m with the belt transect method or with timed-swim surveys. The belt transect method is commonly used for surveying impacts on benthic communities (*Hill & Wilkinson, 2004*). A transect tape deployed along a defined length and within a defined belt at each side of the transect tape. Within this belt, the entire area will be surveyed. The timed-swim surveys are used to observe a large area in a defined time by *e.g.*, snorkeling (*Hill & Wilkinson, 2004*). With this method, the observer swims in zig-zag covering the entire area.

In this study, surveyed reef areas covered between 2,500 to 6,000 m² for each location. The geographic positions of the survey areas were marked with GPS coordinates, but no permanent transects were installed. This means, that the same area was observed between the years and the study hereby provides an overview of the *Xestospongia* population dynamics at all four locations, but there are no statements about the temporal development of single sponge individuals during the survey years.

For the belt transect method, the transects were set parallel to the coastline holding the same water depth of 4 and 6 m. Transect were 50 m in length and 5 m in width to each side resulting in an observed area of 500 m² per transect. In both observation depths, three to six transects were rolled out with a distance of approximately 5 m to each other. In more detail, during all three surveys in Haad Khom (AI) (March 2015, April 2015, 2016), six transects were conducted at each depth (4 and 6 m) covering in total 6,000 m². Due to time constraints on site, both locations in the Marine Park were observed respectively with three transects at each depth (4 and 6 m) during the surveys in March 2015 covering 3,000 m². A second survey at the Marine Park sites in April 2015 was not initiated, because there were no signs of bleaching. During the 2016 surveys in the Marine Park three transects were accomplished at 4 depth but only two transects at 6 m in each location (due to bad weather), covering 2,500 m².

Timed-swim surveys were conducted at Mae Haad (AI) in March 2015 and in 2016 because *Xestospongia* individuals were only found in small, clustered areas during preliminary surveys. The belt transect method was not suitable for this situation, whereas the timed-swim surveys allowed monitoring of a larger area (*Hill & Wilkinson, 2004*). The surveys were conducted by one observer snorkeling, in zig-zag and without gaps, from the shallow (4 m) to the deep (6 m). The observation time was 60 min, resulting in an observed area of approximately 6,000 m² per year. Due to time constraints during the surveys in April 2015, the belt transect method was used with three transects per depth (50 m × 5 m width to each side in 4 and 6 m) at Mae Haad (AI), covering an area of 3,000 m².

All *Xestospongia* individuals within the transects or observed during the timed-swim surveys were counted and measured to the nearest cm (see below) to determine the abundances and sizes. During all surveys, the observer tried their best to document all *Xestospongia* individuals, including very small sizes. Genetic studies indicated that sympatric specimen of *Xestospongia* can belong to a complex of cryptic species that are morphologically not distinguishable. (*Swierts et al., 2017*). Therefore, we refer to all *Xestospongia* specimens as *Xestospongia* sp.

## Population densities and health status of *Xestospongia* sp.

*Xestospongia* abundance was determined by counting all observed individuals. The health status of every individual was documented and assigned to different states: "healthy" and "bleached". "Healthy" means no signs of bleaching or other diseases, and "bleached" characterized individuals with first signs of bleaching up to fully bleached and disintegrated sponges. By bleaching we refer only to the whitish color of the affected specimens, which does not necessarily infer a loss of cyanobacterial symbionts, but rather a diseased state.

## Size distribution of *Xestospongia* sp.

The calculation of the *Xestospongia* biomass was based on different size measurements and corresponding volume concentrations. In detail, the height (h), the osculum diameter (od), and the base diameter (bd) of the sponges were measured in the field. The volume was

afterwards calculated following *McMurray, Blum & Pawlik (2008)*. In this study, sponges were classified into six volume classes to simplify further presentations of the results. Size classes were determined based on noticeable changes in the volume distribution of all investigated sponges in March 2015. The following six volume classes were chosen for this purpose: (I) 0–300 cm³, (II) 301–1,300 cm³, (III) 1,301–10,000 cm³, (IV) 10,001–30,000 cm³, (V) 30,001–100,000 cm³, and (VI) 100,001–300,000 cm³. To extrapolate *Xestospongia* size-at-age, the measured size (volume) data were analyzed using the averaged model from *McGrath et al. (2018)*, who analyzed the growth of Indonesian *Xestospongia* sp.

## Statistics

To compare sponge abundances and the ratio of healthy to bleached sponges between survey depths (4 and 6 m), data from all sites were analyzed separately using Welch's t-test to account for the potential lack of homogeneity (*Rasch, Kubinger & Moder, 2011*). Since depth had no effect on abundance and sponge health, data from both depths were pooled for further analyses. To analyze the effect of the bleaching event on sponge abundances, data from March 2015 and 2016 were compared using a one-way repeated measures ANOVA. Abundance data from Mae Haad (AI) could not be statistically analyzed due to using the timed-swim method with no replication, but with a large observation area in March 2015 and 2016. Statistical analyses were performed with the program SPSS (version 25).

## Sea surface temperature (SST) analyses

To investigate the occurrence of marine heatwaves as possible causes of adverse effects on the health of *Xestospongia* sp. SST data were analyzed. The daily gridded Optimum Interpolation Sea Surface Temperature (OISST) from the "ncdcOisst21Agg_LonPM180" dataset was downloaded (https://coastwatch.pfeg.noaa.gov/erddap/) and prepared with R version 4.3.1 (*R Core Team, 2023*) following https://robwschlegel.github.io/heatwaveR/articles/OISST_preparation.html (at August 7, 2023). The OISST time series was set as climatology period from beginning of January 1987 until the end of December 2016. The most appropriate 0.25° grid cell (9.875°N 99.875°E) was selected, that includes Mae Haad (AI) and is nearby Haad Khom (AI), and analyzed using the "heatwaveR" package version 0.4.6 by *Schlegel & Smit (2018)*. Analyses were conducted according to the instructions by https://robwschlegel.github.io/heatwaveR/articles/event_categories.html and included the marine heatwave definition by *Hobday et al. (2016)* and the nomenclature and categorization by *Hobday et al. (2018)*. Marine heatwaves from the beginning of February 2015 to the end of May 2016 were the focus of analysis.

## RESULTS

### Population densities and health status of *Xestospongia* sp.

The study was initiated in 2015 to assess the abundance and the size distribution of *Xestospongia* sp. in the lower Gulf of Thailand. During the surveys, a mass mortality event of benthic organisms and fish was recognized at the study sites Mae Haad (AI) and Haad Khom (AI), including *Xestospongia* species, the sea snail *Tectus* sp., and bleached

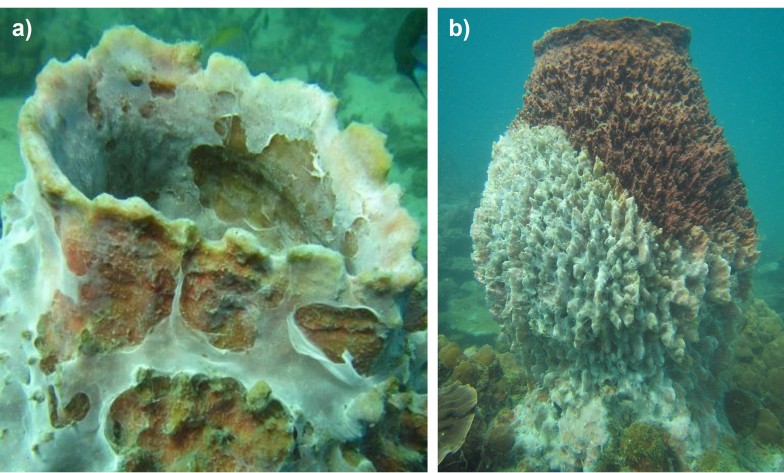

**Figure 2 Impacted _Xestospongia_ sp.** Examples of observed _Xestospongia_ individuals with (A) mucus-like covering and (B) bleaching signs during the studies in April 2015 at Haad Khom (AI) and Mae Haad (AI).

scleractinian corals (_e.g._, _Pocillopora_ and _Acropora_). At the beginning of the observations on 31 March, several healthy-looking _Xestospongia_ individuals with a lighter coloration (bleaching) followed by affected parts being covered with white mucus-like mats were recognized (termed bleached specimens throughout the manuscript, Fig. 2A). During the next days, mucus-like mats expanded on affected specimens and sponge tissue below the mats exhibited lighter coloration compared to non-affected sponge parts (Fig. 2B). Affected sponge tissue disintegrated further over the next days, leading to the collapse of the sponges. Numbers of other affected reef fauna also seemed to increase in the days following the first observation.

During all surveys, mean _Xestospongia_ abundances showed no significant differences (all $p > 0.05$, see Table S1) between the survey depths of 4 and 6 m, except during the surveys at Koh Yippon (ANI) in March 2015, where abundance was higher at 4 m ($p = 0.009$) (see Table S1). Abundance data from Mae Haad (AI) could not be analyzed for differences in depth, due to the adjusted survey method (timed-swim) in March 2015 and in 2016. However, bleaching in April 2015 occurred regardless of the depth in Mae Haad (AI) ($p = 0.179$) and Haad Khom (AI) ($p = 0.150$) (Table S1).

Before the mass mortality event started, 0.02 healthy _Xestospongia_ individuals per m² and no bleached ones were found at Mae Haad (AI) in March 2015 (Fig. 3A). During the second survey (April 2015), 75% of all observed _Xestospongia_ sp. were bleached. One year later, only 0.0003 individuals per m², both healthy-looking, were found. The number of individuals decreased by 98.4% in one year (from March 2015 to 2016).

A similar situation was documented at Haad Khom (AI) (Fig. 3B). During the first survey in March 2015, on average 0.02 healthy-looking _Xestospongia_ individuals per m² and no bleached sponges were counted per transect. During the second survey four weeks later (April 2015), the number of healthy sponges decreased with 83.6% of all observed _Xestospongia_ sp. being bleached. In 2016, on average only 0.004 individuals per m² were

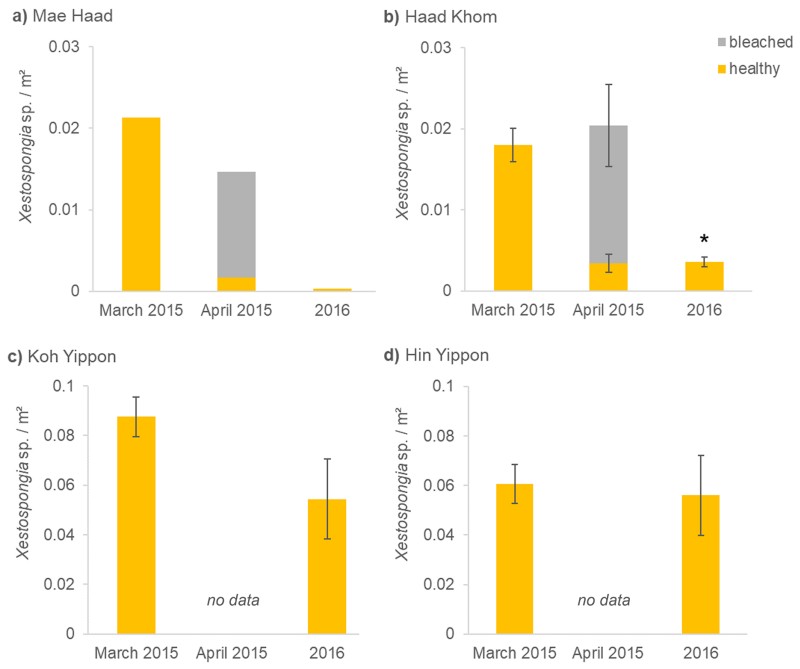

**Figure 3 (A–D)** *Xestospongia* **sp. numbers and fitness status from all locations in the years 2015 and 2016.** The x-axis shows the time of the surveys, separated in pre- (March 2015), peri- (April 2015), and post- mass mortality event (2016). For all locations, the sponge numbers per m² are shown. Different bar colors represent the fitness status: healthy (yellow) and bleached (grey). The standard error of the mean *Xestospongia* counts per transect is given with error bars. The asterisk (*) indicates a significant difference from March 2015 to 2016 with $p < 0.05$ (repeated measures ANOVA).



documented, all looking healthy. The abundance of *Xestospongia* sp. had significantly ($p = 0.000007$) dropped by 80.6% from March 2015 to 2016.

The investigations on the study sites in the Marine Park (ANI) showed no decrease in sponge abundance at Hin Yippon (ANI) ($p=0.3871$), but a decrease from surveys in March 2015 to 2016 at Koh Yippon (ANI) ($p = 0.0563$) (Fig. 3C). The second survey (April 2015) was not conducted at these locations, because there were no signs of bleaching. At Koh Yippon (ANI), the survey in March 2015 showed on average 0.09 *Xestospongia* sp. per m², while sponge counts in 2016 were about 1/3 lower compared to March 2015 (on average 0.05 sponges per m²). All sponges appeared healthy during both surveys. At Hin Yippon (ANI), on average 0.060 *Xestospongia* sp. individuals per m² were documented in March 2015, while 0.056 healthy individuals per m² were documented on average in 2016 (Fig. 3D).

## Size distribution of *Xestospongia* sp.

At Mae Haad (AI), the observed *Xestospongia* individuals were represented in all volume classes during the surveys in March 2015 (Fig. 4A). The class 1,301–10,000 cm³ was thereby most common (40 individuals). The fewest sponge numbers were found in class 100,001–300,000 cm³ (four individuals). In the 2016 surveys, only two sponges in total were documented, both in class 1,301–10,000 cm³.

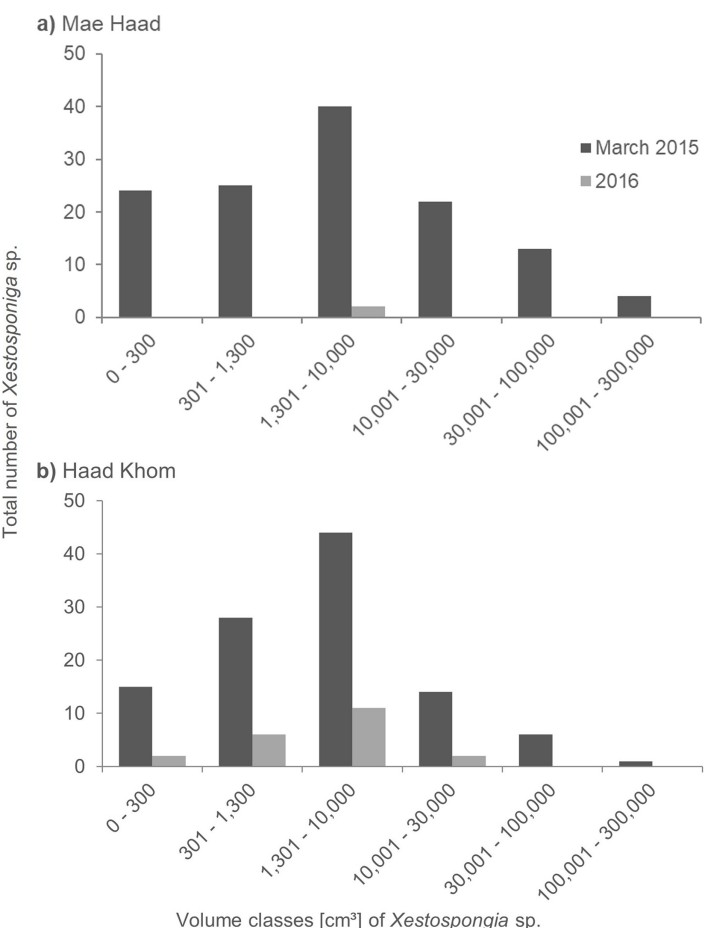

**Figure 4 Size distribution of *Xestospongia* sp. at Mae Haad (AI) and Haad Khom (AI) in the years 2015 and 2016.** The x-axis shows the sponge volume classes, and the y-axis shows the total number of sponges at an area of 6,000 m² in Haad Khom (B) and Mae Haad (A). Size distribution is shown for the surveys in March 2015 (pre-mortality in dark grey) and 2016 (post-mortality in light grey).

At Haad Khom (AI), *Xestospongia* individuals were observed in all volume classes in March 2015 (Fig. 4B). With 44 individuals, most sponges had a volume of 1,301–10,000 cm³, followed by 28 individuals in volume class 301–1,300 cm³. The highest volume class (100,001–300,000 cm³) was represented by a single sponge. In 2016, sponges were just observed in the four first volume classes, the last two classes were not found anymore. Most of the surviving sponges were in the classes 301–1,300 cm³ and 1,301–10,000 cm³ (six and 11 sponges in total). The volume classes 0–300 cm³ and 10,001–30,000 cm³ included two sponges each.

The sponges at the Koh Yippon (ANI) reef were present in all volume classes, and volume distribution was similar in both survey years (Fig. 5A). The volume class 1,301–10,000 cm³ had the highest number of individuals with 102 sponges in March 2015 and 48 sponges in 2016. Volume classes 301–1,300 cm³ and 10,001–30,000 cm³ contained 54 and 51 individuals, respectively in March 2015. The other classes had lower sponge

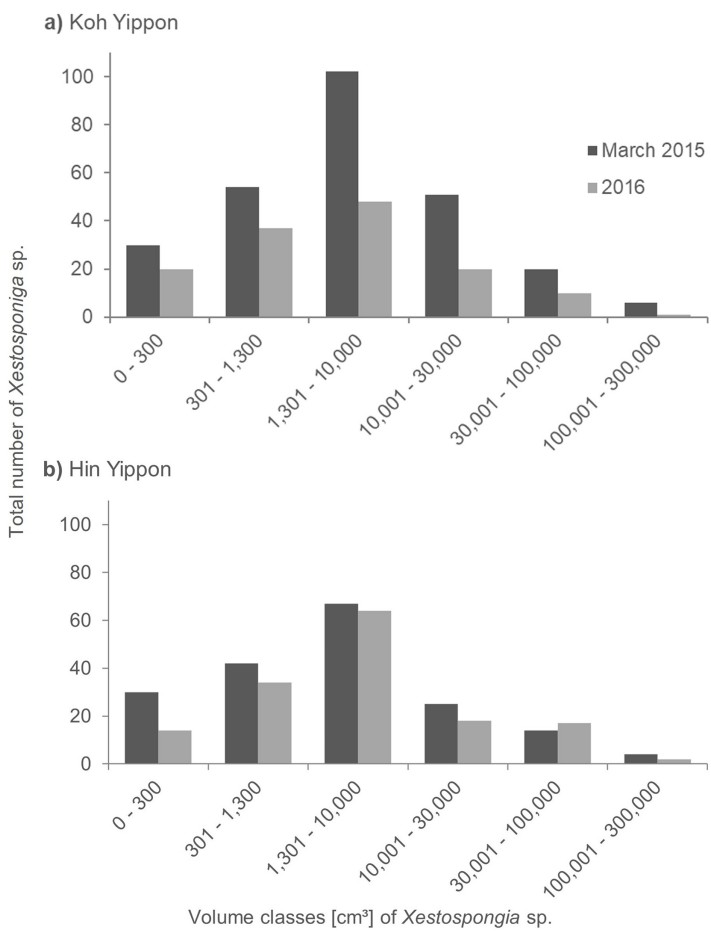

a) Koh Yippon

b) Hin Yippon

Volume classes [cm³] of *Xestospongia* sp.

**Figure 5 Size distribution of *Xestospongia* sp. at Koh Yippon (ANI) and Hin Yippon (ANI) in the years 2015 and 2016.** The x-axis shows the sponge volume classes, and the y-axis shows the total number of sponges in an area of 3,000 m² (2,500 m² in 2016) in Koh Yippon (A) and Hin Yippon (B). Size distribution is shown for the surveys in March 2015 (pre-mortality in dark grey) and 2016 (post-mortality in light grey).

numbers, while the class with volume 100,001–300,000 cm³ had the lowest individual count with 6 *Xestospongia* sp. in March 2015 and a single sponge in 2016.

In Hin Yippon (ANI), every volume class was also represented in both survey years (Fig. 5B). During the surveys in March 2015, the highest number was observed in volume class 1,301–10,000 cm³ with 67 individuals, followed by 42 individuals in volume class 301–1,300 cm³. With four individuals, class 100,001–300,000 cm³ was the one with the lowest *Xestospongia* number. In 2016, this class included two individuals and was therefore also the class with the lowest individual count. Volume class 1,301–10,000 cm³ was with 64 *Xestospongia* individuals the class with the highest count in 2016, followed by volume class 301–1,300 cm³ with 34 sponges.

Further, the weighted average of different growth models was applied to extrapolate the age of the observed sponges (*McGrath et al., 2018*). Sponges of the highest volume class (100,001–300,000 cm³) represent ages of 17–26 years (Fig. S1). The smallest volume class (0–300 cm³) covers the ages 0 up to 1 year. The oldest sponge was estimated to be 24 years

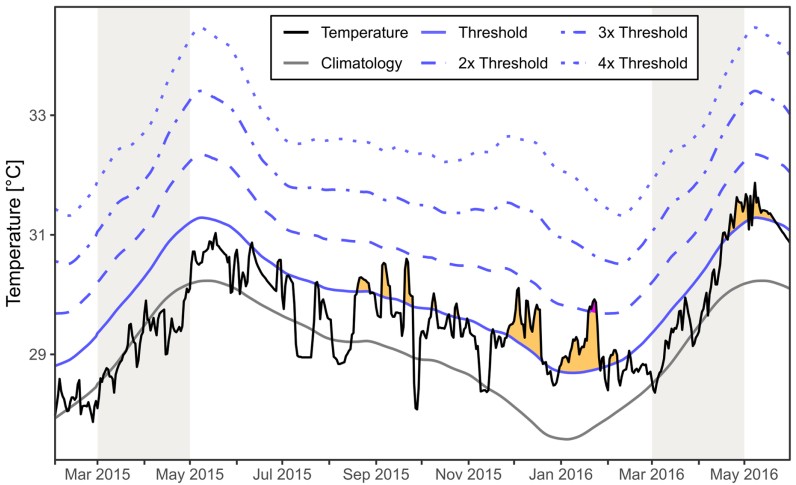

**Figure 6 Regional SST and the occurrence of marine heatwaves.** The SST at Mae Haad (AI) and nearby Haad Khom (AI) (black line) is plotted over the time from February 2015 to the end of May 2016. The plot includes the local climatology (grey line), the 90th percentile climatology (blue line) and its multiples (by blue dashed lines). Yellow areas indicate a moderate marine heatwave and purple a strong marine heatwave (occurred only end of January 2016) following definitions by *Hobday et al. (2018)*. Grey colored areas indicate the months of fieldwork.

old and was found during the surveys in March 2015 in Haad Khom (AI). This sponge had a volume of 236,544 cm³.

## Sea surface temperatures

The regional SST for the area around Mae Haad (AI) and nearby Haad Khom (AI) indicated no marine heatwaves during the observed mass mortality of *Xestospongia* sp. between February and April 2015 (Fig. 6). Moderate marine heatwaves were found irregularly after the investigations in 2015 and before fieldwork in 2016, as well as at the end and after the months of investigation in 2016. Furthermore, a strong marine heatwave occurred in January 2016.

## DISCUSSION

This study documents a locally restricted massive die-off of the sponge *Xestospongia* sp. While the northwestern reefs of Koh Phangan, Mae Haad (AI) and Haad Khom (AI) were subject to a population loss of 75–83%, the close by reefs in the Mu Ko Ang Thong Marine National were not affected by this phenomenon. The massive mortality occurred regardless of the survey depth (Table S1), leading to a rapid decline in sponge abundances by 98.4% at Mae Haad (AI) and 80.6% at Haad Khom (AI) from year 2015 to year 2016.

Affected sponges showed a decrease in their usual brown pigmentation with these "bleached" parts becoming covered by whitish to grey mats within a day or two. Once the sponges were completely covered by these presumably microbial mats, the sponge tissue disintegrated further leading to the collapse of especially large *Xestospongia* individuals. Disintegration of the tissue continued until only bare skeletal fibers of dead sponges were left, making dead sponges difficult to detect during surveys. Based on our observations it cannot be determined if the microbial mats were the cause or rather a secondary infection
of opportunistic microorganisms, following the initial bleaching of affected sponges. *Ereskovsky et al. (2019)* described a similar mass mortality in several arctic sponges following an abnormal heatwave in the area. Initial infected parts showed discoloration followed by tissue necrosis and further loss of natural coloration. In some species the tissue became also covered by mucus mats. Similar to our study they could not determine the cause of the mortality event, whether the observed heatwave directly led to the mass mortality or if it acted in synergy with waterborne agents.

Although a heatwave could not be confirmed in our case from historical data, affected *Xestospongia* sp. sponges revealed similar disease progression leading to mortality as described by *Ereskovsky et al. (2019)*.

Based on *McMurray et al. (2011)*, fatal bleaching of *Xestospongia* sp. in the Caribbean usually does not result from cyclic bleaching, but can be caused by *e.g.*, diseases and other pathogens. According to our observations, the mortality event in April 2015 was not a specific sponge disease as numerous *Xestospongia* individuals, other sponge species and corals adjacent to affected sponges started to bleach. During the surveys, the mortality of further marine fauna was documented, including numerous individuals of the marine snail *Tectus* sp., fish, and crustaceans (P-J Grammel and E Schoenig, 2015, personal communication). Taken these observations into account, the mass mortality event could have had various reasons. Having numerous other species from different phyla affected, could most likely indicate three possible scenarios for this mortality event: local higher water temperatures, a local harmful algal bloom (HAB), or a combination of both factors.

Mass mortality events affecting species from various phyla have been reported from the Mediterranean Sea as a result of marine heatwaves with increased sea water temperatures (*Garrabou et al., 2022*, *2009*; *Rubio-Portillo et al., 2016*). Studies from other locations have reported increased water temperatures due to El Nino Southern Oscillation in 2015/2016, which affected marine organisms and resulted likewise in bleaching and mortality (*Head et al., 2019*; *Rubio-Portillo et al., 2016*). *Strano et al. (2022)* could also demonstrate in laboratory experiments with the temperate model sponge *Crella incrustans*, that experimentally induced heatwaves led to disease and mortality in the exposed sponges. Sponge mortality events have also been reported by *Stevely et al. (2011)* from the Florida Keys. They observed widespread sponge mortality of various sponge species which occurred together with cyanobacterial blooms, the likely cause of the observed mortality. They also documented long-lasting ecological consequences for the recovery of sponge communities for more than one decade.

There was no regional marine heatwave detected during the observed *Xestospongia* sp. mass mortality event between February and April 2015 based on the SST satellite dataset for the area around Mae Haad (AI) and nearby Haad Khom (AI). Therefore, it is highly unlikely that a marine heatwave directly caused the observed mass mortality in April 2015. However, heatwaves after the investigation in April 2015 could have reduced the chance for recovery or could have increased further mortality of *Xestospongia* sp. by April 2016.

The mass mortality event was only a local phenomenon. Only sponges at the two monitored reefs of Koh Phangan were impacted, while no affected sponges were found at the same water depths at the monitored reefs of Koh Yippon (ANI) and Hin Yippon

(ANI). Considering these circumstances, it becomes most likely that the mass mortality event may have been triggered by a local HAB.

In general, HAB events are increasing rapidly in density, distribution, and intensity, and most coasts worldwide are nowadays threatened (*Bauman et al., 2010*; *Gobler et al., 2017*; *Xiao et al., 2019*). This is caused amongst others by climate change and increasing nutrient pollution (*Gobler, 2020*; *Heisler et al., 2008*). HABs are caused by toxic microalgae or cyanobacteria, which can lead during outbreaks to illness and death of fish, other marine invertebrates, seabirds, and even humans (*Starr et al., 2017*; *Young et al., 2020*). Sponges are important for benthic-pelagic coupling as they filter large amounts of water (up to 1,000 $L^*h^{-1}*kg^{-1}$ body mass, (*Vogel, 1977*)). This in turn could make them also very susceptible to HABs, by retaining the microalgae and thereby accumulating the produced toxins, leading ultimately to the death of the affected sponges (*Anderson, 2007*). In 2015, HAB events were reported for other regions around the world, including the U.S. West Coast, coastal areas in Northern Europe, and Canada (*Karlson et al., 2021*; *McCabe et al., 2016*; *McKenzie et al., 2021*). For example, the HAB at the U.S. West Coast was a coastwide event, negatively impacting fisheries and initiated by anomalously warm water conditions (*McCabe et al., 2016*; *Moore et al., 2020*).

At our study site in Mae Haad (AI), *Stuhldreier et al. (2015)* reported high phosphate concentrations in a small river carrying untreated sewage water from nearby bungalows, which drained into the bay through small water channels on the beach. Both, the reported increased nutrient pollution by the river and effects of El Nino Southern Oscillation, could be possible triggers for a HAB outbreak resulting in the observed mass mortality of *Xestospongia* sponges at the study sites Mae Haad (AI) and Haad Khom (AI). But it is difficult to determine the reason for the sponge mortality event because environmental parameters such as nutrients, pH, salinity were not taken, due to a lack of analytical capabilities in 2015. Therefore, we can only speculate on the cause of this mortality event.

*Rosenfeld (2021)* compared different environmental parameters around five sites on Koh Phangan in 2019, including reefs in Mae Haad (AI) and Haad Khom (AI). Differences in environmental parameters (pH, NOx, and $PO_4^{3-}$) were rather small between the bays ranging in pH from 7.95 to 8.12, in NOx from 2.44 to 2.91 µmol $L^{-1}$, and for $PO_4^{3-}$ from 1.52 to 1.73 µmol $L^{-1}$. However, analysis of the Mae Haad river, which discharges into Mae Haad bay (AI) showed very high nutrient levels with 7.24 µmol $L^{-1}$ for NOx and 6.06 µmol $L^{-1}$ for phosphate. Such high nutrient levels could certainly promote algae blooms at Mae Haad (AI), although we do not know if such high nutrient levels were also present in 2015 during the mass mortality event. Monitoring programs and management actions in Koh Phangan are needed in the future to detect possible (human) impacts and to reduce negative consequences on ecosystem health.

The reefs of Koh Yippon (ANI) and Hin Yippon (ANI) showed just little changes in the abundance of *Xestospongia* sp. between the years 2015 and 2016. These small variances could be either caused by growth effects over the course of one year, or by monitoring artefacts, since the position and the number of the transect tapes differed between the two survey years. However, these anthropogenic non-impacted study sites represented in both survey years populations with a higher number of observed *Xestospongia* sp. than at the

anthropogenically impacted locations. A decline in sponge health was not detected during the surveys at both Marine Park locations.

Regarding the size distribution of *Xestospongia* sp., sponge sizes (according to volume classes) showed a similar Gaussian distribution at all study sites in both survey years. The highest number of *Xestospongia* sp. were found for all study sites at the middle class 1,301–10,000 cm³. The lower number of small individuals could possibly be caused by a higher risk of being eaten by predators. Larger individuals are probably more often affected by environmental factors over time due to their higher age (*Smith & Smith, 2009*). The mass mortality event also showed its effects on the size distribution. Only two sponges of the previously dominant middle class were still alive at Mae Haad (AI) in 2016. At Haad Khom (AI), *Xestospongia* individuals within the small and middle classes survived after the massive bleaching. The results show that sponges of all sizes were affected by the mass mortality event. This observation suggests an exceptional event occurred in the northern region of Koh Phangan, as large sponges, known for their resilience and ability to withstand cyclic bleaching, disappeared in 2016.

To estimate the age of sponge individuals, size-at-age models have been used in earlier studies. Reports from Caribbean *Xestospongia muta* revealed ages of 127 years, and estimated the largest sponges on these reefs to be more than 2,300 years old, reflecting *Xestospongia* spp. as a long-lived and slow-growing sponge species (*McMurray, Blum & Pawlik, 2008*). Contrary and more recently, *McGrath et al. (2018)* found Indo-Pacific *Xestospongia* spp. from Sulawesi, Indonesia, to be faster growing than their Caribbean relatives. We used the parameter from the growth model from *McGrath et al. (2018)* to estimate the age of our sponges, since they are both Pacific populations and represent more likely the same species. The largest measured individuals of the investigated *Xestospongia* sp. (March 2015 at Haad Khom (AI)) were up to 24 years old (Fig. S1). One year later, the oldest sponge at the same location was only 9 years old. Sponges of the middle volume classes, which were most abundant and survived the bleaching, were around 3–7 years old.

Although three species of *Xestospongia*, namely *X. muta, X. testudinaria*, and *X. bergquistia*, are recognized globally, recent studies suggest that these species are part of complexes consisting of cryptic species (*Evans et al., 2021*; *Swierts et al., 2017*). *Swierts et al. (2017)* analyzed 10 *Xestospongia* samples from Koh Tao (only ~40 km away from our sampling sites) and identified three different haplotypes. However, the limitations of the genetic markers made it impossible to determine the number of species. The current species consensus of all these samples was *X. testudinaria*. Despite being cryptic species and challenging to differentiate, different *Xestospongia* species can exhibit varying growth performances. For instance, *Deignan, Pawlik & López-Legentil (2018)* explored intraspecific variation within a single population of *X. muta* in Florida, and identified two distinct, co-occurring genetic clusters. Furthermore, they found that each genetic cluster of *X. muta* exhibited a preference towards different sponge sizes, highlighting the variation in growth performance among these cryptic species. We can hypothesize that in addition to differences in growth performance among different *Xestospongia* species, these variations may also be attributed to other ecological differences, such as their susceptibility to diseases or heat stress. Dealing with potentially cryptic species, it becomes essential to consider that

factors like disease resistance or tolerance could contribute to the observed dissimilarities in growth patterns. Understanding how various ecological factors interact with genetic variations can provide valuable insights into the complexities of *Xestospongia* populations and their overall ecological dynamics. Further research is necessary to explore these aspects comprehensively and gain a more comprehensive understanding of the factors influencing the growth and ecological dynamics of *Xestospongia* species.

## CONCLUSIONS

This study documented an extraordinary massive mortality event of *Xestospongia* populations from two anthropogenic impacted reefs of Koh Phangan, Thailand. *Xestospongia* sp. mortality ranged from 98.4% at Mae Haad (AI) to 80.6% at Haad Khom (AI) (between monitoring events in 2015 and 2016). Anthropogenic non-impacted reefs at the Mu Ko Ang Thong Marine National Park were not affected. This underlines the importance to establish and enforce protected Marine Parks for maintaining healthy ecosystems. The localized mortality event was probably triggered by a local HAB, pathogens, undetected local higher water temperatures, or a combination of these factors. Whereas no marine heatwave was detected during the observed mass mortality event in 2015 from sea surface temperature data. *Xestospongia* sp. of all sizes were affected. The lack of *Xestospongia* sp. following the mass mortality event in the two surveyed bays, could lead to a decrease in biodiversity, a weaker filter efficiency and loss of 3D complexity and function (*e.g.*, habitat and shelter) of the reefs. The enormous loss of *Xestospongia* sp. in Mae Haad (AI) and Haad Khom (AI) could also have serious long-term consequences for the sponge populations and for the coral reefs in these locations, as reproduction and recruitment patterns for this species are unknown. Therefore, it will be important to determine and reduce triggers of such mass mortality events, especially for essential habitat-forming organisms such as *Xestospongia* spp. Long-term monitoring programs are needed to document the condition of reef systems and its ecological parameters, and to detect and act on the causes of decline.

## ACKNOWLEDGEMENTS

In memory of our colleague Eike Schoenig, who passed away during the writing of this manuscript. As founder of CORE sea, he was a role model of dedication and commitment to the protection of coral reefs. As an adviser and teacher for students, he left a legacy for many to follow his lead. He was a pioneer in marine research of Koh Phangan and essential in conducting this research project and supporting the students working on it. We are very grateful to him for all his support on this study.

The authors also acknowledge Janina Schoenig for her support in student mentoring. We would like to sincerely thank Florian Piehl (ICBM, Germany), the staff, and volunteers at CORE sea (Thailand) for their support during the fieldwork. We also would like to thank the editor and the anonymous reviewers for their comments, which helped to improve the manuscript.

This study is the result of two bachelor theses from the Bachelor's Program "Environmental Sciences" at the University of Oldenburg, submitted by JSM (2016) and PJG (2016).

### Funding
The authors received no funding for this work.

### Competing Interests
The authors declare that they have no competing interests.

### Author Contributions
- Jasmin S. Mueller performed the experiments, analyzed the data, prepared figures and/or tables, authored or reviewed drafts of the article, and approved the final draft.
- Paul-Jannis Grammel performed the experiments, analyzed the data, prepared figures and/or tables, authored or reviewed drafts of the article, and approved the final draft.
- Nicolas Bill analyzed the data, prepared figures and/or tables, authored or reviewed drafts of the article, and approved the final draft.
- Sven Rohde conceived and designed the experiments, authored or reviewed drafts of the article, and approved the final draft.
- Peter J. Schupp performed initial experiments, conceived and designed the experiments, authored or reviewed drafts of the article, and approved the final draft.

### Field Study Permissions
The following information was supplied relating to field study approvals (*i.e.*, approving body and any reference numbers):

Surveys were conducted under a Memorandum of Understanding with the Department of Marine and Coastal Resources Thailand and the Phuket Marine Biological Center.

### Data Availability
The raw data of Xestospongia abundances, the measured values to determine sponge volumes and the calculated volumes, and the fitness status in 2015.2 are available in the Supplemental Files.

### Supplemental Information
Supplemental information for this article can be found online at http://dx.doi.org/10.7717/peerj.16561#supplemental-information.

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
