# Peer review of "Mass mortality event of the giant barrel sponge Xestospongia sp.: population dynamics and size distribution in Koh Phangan, Gulf of Thailand"

_PeerJ, doi:10.7717/peerj.16561_

## Round 0.1 · original submission · Major Revisions

We have received evaluations from three expert reviewers and their comments can be seen below and in a separate document. The reviewers agree that the manuscript provides useful results on an important but understudied ecosystem engineer in a region where little information is available. However, there many issues have been pointed out and these need to be addressed in a revised version of the manuscript

In the introduction, it would be good to include more background information for context about the study area and about the species complex. Also, please provide more information in the Materials and Methods to ensure that it is complete. The discussion needs more information as well.

The text requires a fluent English speaker or a language service to review the scientific and English language. In some parts, the manuscript is repetitive, in other parts references are needed to back up statements. Also, the flow of ideas and arguments needs to be improved throughout the manuscript as does the use of scientific terminology.

If you decide to resubmit the manuscript for further consideration, please ensure that you provide a detailed response to reviewers along with a tracked changes and a clean version of the manuscript.

Reviewer 1 ·

Basic reporting

English is good.

Literature cited is sufficient, but could be expanded with more recent articles on the topic.

Structure, figures, tables and raw data are appropriate

The article is self-contained

Experimental design

Article is original.

Research questions are relevant and meaningful, but are not well defined. The authors should expand the aim of the study in a separate paragraph at the end of the introduction.

The investigation is rigorous and the weaknesses are acknowledged in the discussion.

Methods are detailed and well written

Validity of the findings

All good.

Additional comments

In the paper "Mass mortality event of the giant barrel sponge Xestospongia sp.: Population dynamics and size distribution in Koh Phangan, Gulf of Thailand", Mueller et al. presented the first documentation of a mass mortality event of the giant barrel sponge Xestospongia sp. in the lower Gulf of Thailand and its consequences on population dynamics and size distribution. The authors surveyed two anthropogenic impacted reefs and two anthropogenic non-impacted reefs within the Mu Ko Ang Thong Marine National Park in the years 2015 and 2016. The results showed a strong shift in population densities at Koh Phangan and a significant decrease in Xestospongia sp. abundance due to fatal bleaching resulting in mass mortality in 2015.

I find the study very interesting and important, but the article presents some fundamental issue that need to be addressed before the paper can be considered for publication.

First issue: while I understand that environmental data are not available for the study site during the period investigated, I believe that sea surface temperature data are crucial to understanding the cause of the bleaching event in this study. Such data are readily available online, along with tools for analyzing climatology and identifying possible heatwaves. Upon conducting a preliminary check, I found that SST in the study areas between March and April 2015 was not significantly higher than the average temperature for that period, making the hypothesis of a heatwave causing the bleaching event highly unlikely. I highly recommend that the authors formally analyse SST data for the areas investigated and discuss the results. The standard method for analyzing sea surface temperature is currently described in Hobday et al. (2016, https://doi.org/10.1016/j.pocean.2015.12.014). The authors may refer to the instructions for downloading NOAA daily data and detecting heatwaves provided at this link: https://robwschlegel.github.io/heatwaveR/articles/OISST_preparation.html.

My second major comment concerns the authors' use of the term "bleaching." Based on the photos provided, it appears that the authors may be dealing more with bacterial infection following necrosis/dysbiosis than with bleaching caused by symbiont loss. The white mucus-like film shown in Figure 2A is most likely bacteria, and while it is difficult to discern from the low-resolution photos in Figure 2B, the white on the sponges also appears to be due to the same bacterial film. This phenomenon is often observed in stressed sponges before or during their death or tissue necrosis, and is very different from bleaching. In cases of bleaching, the sponge tissue often appears healthy and there is no associated bacterial film. Given that all the sponges died after the event, it is possible that the authors were actually dealing with necrosis rather than bleaching. Moreover, bleaching is mostly associated with increased temperature, and as per my previous comment, it does not seem that temperature was the problem in this case. I would strongly urge the authors to carefully investigate whether the "white stuff" they observed was indeed due to the absence of symbionts (i.e., bleaching) or due to bacterial film (i.e., necrosis).
A couple of useful papers on the matter are below:
Ereskovsky et al. 2012. Polar Biology. https://link.springer.com/article/10.1007/s00300-019-02606-0
Strano et al. 2022. Science of the Total Environment. https://doi.org/10.1016/j.scitotenv.2022.153466

Some other minor comments:

Line 46: Provide -> Providing
Line 91: Please also look at the new review on the shift from corals to other animal-dominated reefs: Bell et al. 2022. https://doi.org/10.1042/ETLS20210231
Line 235 : What “both” is referred to?
Discussion: lease consider splitting the first part of the discussion into different paragraphs. With many shorter paraphraphs, it will make it easier for readers to find the information they are looking for.

In the text and in the figures, please replace 2015.1 and 2015.2 with something easier to understand, at least in the discussion, like March 2015 and April 2015. It is easy to forget what .1 .2 mean.

Reviewer 2 ·

Basic reporting

I general, texts are clear (good English) and brig sufficient field background/context.

Below, some questions, missing items and suggestions:

• Line 72. “spp.” after a genus means more than one species, i.e., in plural. Then, should the phrase mention about the genus or species of Xestospongia? Review it!
• Line 80. “However, McGrath and colleagues (2018) found contrary to…” would be a new paragraph, but cut off “However,”.
• Line 94. It is not a new paragraph, it should be continued on line 93.
• Line 104. It is a new paragraph.

Experimental design

All items were achieved. However, Field survey methods from M&M. It is a little confuse. Should be it clearer here. I think should you, in first time, only describe the methods (belt transect and time-swim surveys), then mention surveyed area and all other related things about them, employed method(s) to each one, year surveyed, xx m2 covered, etc.

Below, some questions, missing items and suggestions:

• Line 121. “the sponges are further referred to as Xestospongia sp.”, are you mentioning the material analyzed? Make it clear.
• Line 153. Xestospongia is not in italic format.
• Line 164. Mention 3,000 m3, but should be 3,000 m2.

Validity of the findings

Impact and novelty assessed from the survey.

Data
• they are provided and satisfactory for the study


Conclusions
• Lines 409 to 410. Mention that localized mortality event was probably triggered by a local HAB or undetected local higher water temperatures. I think that you should mention a possible combination of both factors too, since this idea has been mentioned in the Discussion.
• Line 412. McMurray et al. (2014, 2010) have been cited, but citations should not be presented in the Conclusion. You should just present the main conclusions.

Additional comments

Since specimens of Xestospongia from the present study are not identified (species level) would be interesting discuss about this. The authors mention in the introduction that Xestospongia specimens form Thailand is part of a species complex., then would there is a possibility of more than one species from the present study. How this affect the outcomes of the present survey? I believe that this should be discussed.


The manuscript about “Mass mortality event of the giant barrel sponge Xestospongia sp.: Population dynamics and size distribution in Koh Phangan, Gulf of Thailand” is a good study/survey and it is suitable for publication on PeerJ. However, there are some a little adjustments to a final version of this manuscript.

Below, some questions, missing items and suggestions:

• Lines 293 to 296. Sound as Results. Cut off, please!
• Lines 297 to 299. Bare skeletal fibers of dead sponges are mentioned and soon after was mentioned the bleaching and mortality from the present study. So, how are both ideas related one each other? This paragraph need improvement.
• Lines 299 to 230. Be clearer! Here you can mention the surveyed sites where there were rapid decline of Xestospongia abundance. So, here you can mention again percentages per sites.
• Line 301. It should be a new paragraph.
• Line 335. Started with “At our study site in Mae Haad, Stuhldreier et al. (2015)…”, it should be a new paragraph.
• Lines 336 to 337. Who have show high phosphate concentrations after subsequent analyses of the river water? Be clearer, please.
• Line 340. Mention here the surveyed sites where were observed mass mortality of Xestospongia sponges.
• Line 343. Started with “Rosenfeld (2021, unpublished data) compared...”, it would be a new paragraph.
• Line 359. Started with “However, the anthropogenic non-impacted reefs…”, cut off “However” and start with “The anthropogenic non-impacted reefs…”.
• Line 361. Did you mean “low” variance?
• Lines 367 to 369. Started with “These facts lead to the assumption…”, it sound as Conclusions.
Conclusions
• Lines 409 to 410. Mention that localized mortality event was probably triggered by a local HAB or undetected local higher water temperatures. I think that you should mention a possible combination of both factors too, since this idea has been mentioned in the Discussion.
• Line 412. McMurray et al. (2014, 2010) have been cited, but citations should not be presented in the Conclusion. You should just present the main conclusions.

Reviewer 3 ·

Basic reporting

The submitted manuscript represents an important report of a massive mortality event of a conspicuous sponge in the Gulf of Thailand, showing the importance of constant monitoring of reef environments and the establishment of Marine Protected Areas.
However, I consider the text needs great improvement concerning the English language and towards a more professional, scientific, and less colloquial writing (especially in the introduction and discussion). I suggest revising it extensively (I give some specific examples in the attached pdf).
Also, I think the authors should improve the background of the sampling sites and, especially of the focal species on the study area, in the introduction, exploiting and presenting more clearly the current knowledge concerning these topics to improve the understanding, importance and context of the work.
I consider the presented figures are relevant, although I think the map could be more informative (see pdf) and needs improvement in resolution. Raw data were available and clear, in my opinion.

Experimental design

I consider that the manuscript fits the Aims and Scope of PeerJ and the experimental design is suitable for the proposed objectives.
However, I think the authors could invest more time to better explain some small issues in the Materials and Methods, to make its description clearer (I give more specific examples in the attached pdf).

Validity of the findings

I consider that the manuscript brings important results of a massive mortality event of a conspicuous and important ecosystem engineer sponge species. More importantly, it reports this event in a poorly studied area, showing the effects of human activities, and the importance of reef environment monitoring and establishment of Marine Protected Areas.
However, I think the authors could search for specific environmental data and statistics that could help explain or corroborate their observations, improving and making the discussion less speculative. Also, I think its important to review the text of the discussion to improve the English, as it has some parts with colloquial terms and not with a standard scientific language.

Additional comments

Please, see the attached pdf for more comments

Annotated reviews are not available for download in order to protect the identity of reviewers who chose to remain anonymous.

---

## Round 0.2 · Minor Revisions

We have received evaluations back from two expert reviewers. Both remark on the marked improvement to the manuscript. One reviewer has suggested a few minor clarifications with regard to the SST data and their analysis. I agree with the reviewer that these should be attended to in a revised version of the manuscript.

Reviewer 1 ·

Basic reporting

all good

Experimental design

The authors have adequately addressed previous concerns and the manuscript has improved substantially. I appreciate the inclusion of SST data analysis to investigate potential thermal anomalies related to the mass mortality event. For clarity, I recommend the following minor addition:

In the Materials and Methods, briefly describe the SST data utilized and methods for analysis. In the Results, include a figure showing the SST climatology for the time period of the mortality event (using the package HeatwaveR, https://robwschlegel.github.io/heatwaveR/articles/event_categories.html). Though no thermal anomaly was found, visually representing the data better supports this conclusion. In the Discussion, interpret the SST findings in context of the proposed heatwave hypothesis.

Overall the manuscript is much improved and I recommend publication pending the inclusion of the SST data analysis details suggested above. This will improve transparency and allow readers to better evaluate the heatwave hypothesis.

Validity of the findings

all good

Reviewer 3 ·

Basic reporting

I think the authors fully addressed all the concerns pointed along the text by the reviewers. Therefore, now I consider that the manuscript is suitable for publication in PeerJ.

Experimental design

No extra comments

Validity of the findings

No extra comments

Additional comments

I would like to thank the authors for being so polite and open about the topics pointed by the reviewers, accepting all the suggestions. Congratulations

---

## Round 0.3 · accepted · Accept

I am very pleased to let you know that the modifications that you made to the manuscript have been found to be acceptable and there are no further issues. I recommend that this manuscript be accepted for publication. Congratulations.

Reviewer 1 ·

Basic reporting

All good now, I recommend the article for publication.

Experimental design

All good

Validity of the findings

All good